# Prevalence and Type Distribution of High-Risk Human Papillomavirus (HPV) in Breast Cancer: A Qatar Based Study

**DOI:** 10.3390/cancers12061528

**Published:** 2020-06-10

**Authors:** Gulab Sher, Nadia Aziz Salman, Michal Kulinski, Rayyan Abdulaziz Fadel, Vinod Kumar Gupta, Ambika Anand, Salahddin Gehani, Sheraz Abayazeed, Omer Al-Yahri, Fakhar Shahid, Salman Alshaibani, Sara Hassan, M. Zafar Chawdhery, Giles Davies, Said Dermime, Shahab Uddin, G. Hossein Ashrafi, Kulsoom Junejo

**Affiliations:** 1Translational Research Institute, Academic Health System, Hamad Medical Corporation, Doha 3050, Qatar; GSher@hamad.qa (G.S.); MKulinski@hamad.qa (M.K.); VKumar3@hamad.qa (V.K.G.); SKhan34@hamad.qa (S.U.); 2School of Life Science, Pharmacy and Chemistry, Kingston University London, London KT1 2EE, UK; k1246161@kingston.ac.uk; 3Breast Cancer Unit, Hamad General Hospital, Hamad Medical Corporation, Doha 3050, Qatar; brownyf8@gmail.com (R.A.F.); aanand@hamad.qa (A.A.); sgehani@hamad.qa (S.G.); sahmed42@hamad.qa (S.A.); oalyahri@hamad.qa (O.A.-Y.); fshahid@hamad.qa (F.S.); salshaibani@hamad.qa (S.A.); shassan16@hamad.qa (S.H.); 4Surgery Department, St. Anthony’s Hospital, Surrey SM3 9DW, UK; mzc@doctors.org.uk; 5Breast Surgery Department, BUPA Cromwell Hospital, London SW5 0T, UK; GilesDavies@thebreastclinic.co.uk; 6National Center for Cancer Care and Research, Hamad Medical Corporation, Doha 3050, Qatar; SDermime@hamad.qa

**Keywords:** breast cancer, HPV, PCR, prevalence, genotype, Qatar

## Abstract

Human papillomavirus (HPV) has been implicated in the etiology of a variety of human cancers. Studies investigating the presence of high-risk (HR) HPV in breast tissue have generated considerable controversy over its role as a potential risk factor for breast cancer (BC). This is the first investigation reporting the prevalence and type distribution of high-risk HPV infection in breast tissue in the population of Qatar. A prospective comparison blind research study herein reconnoitered the presence of twelve HR-HPV types’ DNA using multiplex PCR by screening a total of 150 fresh breast tissue specimens. Data obtained shows that HR-HPV types were found in 10% of subjects with breast cancer; of which the presence of HPV was confirmed in 4/33 (12.12%) of invasive carcinomas. These findings, the first reported from the population of Qatar, suggest that the selective presence of HPV in breast tissue is likely to be a related factor in the progression of certain cases of breast cancer.

## 1. Introduction

Breast cancer (BC) is one of the leading causes of cancer death among women worldwide and its burden on women is markedly increasing. According to the World Health Organization, female BC accounts for 626,679 deaths each year worldwide. Since 2008, the BC morbidity rate has increased by more than 20% and the BC mortality rate has jumped by 14% [1]. In Qatar, BC accounts for 39.41% of all female cancers where 242 new breast cancer cases were diagnosed in 2015 with 72.08 per 100,000 of the population at risk of developing the disease [2]. This emphasizes the importance of identifying novel etiological risk factors that are associated with the development of BC.

Infectious agents, such as viruses, are known to play a significant role in cancer initiation. These biological risk factors contribute to approximately 18–20% of human cancers. Human papillomavirus (HPV) is a sexually transmitted virus, which infects mucosal and cutaneous epithelia [3]. HPV infection has been assumed to be initiated mainly by direct skin-to-skin or skin-to-mucosa contact. Studies have shown HPV transmission between the hands and genitals of the same person and/or partners. It has been estimated that 80% of the population, worldwide, has been infected with at least one HPV type at some time during their lifetime [4,5].

High-risk (HR)-HPV accounts for more than 50% of infection-linked cancers in women. The causal link between genital HPV infection and the development of cervical cancer is well-established [6,7]. In the presence of other cofactors, long-term viral infection persistence is required for the initiation and progression of malignancy [8]. Recently, it has been suggested that some viruses, including HPV, are among the potential risk factors for the development of BC. However, the implication of HPV infection being an etiological risk factor for breast cancer carcinogenesis remains controversial [9,10].

The relationship between HPV and BC is imperative for several reasons. The anatomy of the mammary ducts exposes the opened ductal pipelines to the external environment, and this increases the risk of HPV infection. Most breast cancer cases originate from mammary duct epithelia, where ductal hyperproliferation is followed by subsequent tumor progression to in situ and invasive ductal carcinomas [11,12]. Thus, it has been postulated that HPV virions might be transported from the original infection site in the genital area, to enter and infect the mammary ducts through the nipple [13]. Subsequently, this may be responsible for the development of certain types of BC [14].

Several studies have been conducted to establish the presence of HPV in BC as a causal or associated risk factor for particular types of the disease, yet this remains a moot point. In addition, to date, no data is available on the HPV burden in the general population of Qatar. This study is, to the best of our knowledge, the first to investigate the presence of HR-HPV types within the BC of the population of Qatar and then compare it with a different population: the population of the UK.

## 2. Results

### 2.1. Histopathological Diagnosis

The breast tissue biopsy specimens (*n* = 150) were collected and their histopathological diagnoses established, as collated in Table 1. As expected, malignancy was confirmed in 50 out of the 150 breast tissue specimens (15/50 were in situ carcinoma, 34/50 invasive carcinoma and 1/50 was an invasive and in situ carcinoma). Diagnosis of benign disease was reported in 50 specimens (23/50 were benign fibroadenoma, 4/50 benign phyllodes tumor, 13/50 benign breast tissue, 5/50 gynecomastia and 5/50 were papillomatosis). The remainder 50/150 specimens were described as normal breast tissue.

### 2.2. PCR Results

PCR was performed in triplicate for each HPV positive sample to confirm the result. Since a very strict procedure for contamination control was followed in the study, no carryover or failure of the test system was observed according to in-run control results. One hundred and fifty fresh samples from 140 females and 10 males aged 15–84 years were analyzed for the DNA presence of 12 HR-HPV types using type-specific PCR. The amplification of β-globin gene (fragment size of a 723 bp) was positive in all samples, showing an adequate quality of DNA. Positive and negative controls were also run for each sample. As shown in Figure 1, the resulting fragments of HPV type-specific positive controls and negative controls were clarified effectively and no evidence of contamination was found, indicating a successful PCR amplification.

### 2.3. Prevalence and Type Distribution of HR-HPV

The presence of HR-HPV DNA was detected in 13 out of 150 samples (8.7%) (Table 1 and Table 2). In malignant samples, HR-HPV DNA was detected in 5/50 with a prevalence of 10% (Table 1) in which HPV 16 and 35 were the most prevalent HPV genotypes each in 3/50 (6%) samples, followed by type 58 in 1/50 (2%) (Table 1) (Figure 2). In benign/normal samples, HR-HPV DNA showed a prevalence of 8/100 (8%) in which HPV 35 was the most prevalent genotype in 4/100 (4%) followed by type 52, 16, 33, 18, 39, 59 and 58 with respective prevalence of (2%, 1%, 1%, 1%, 1%, 1%, 1%) (Table 1) (Figure 2). The frequency of co-infection with multiple HPV genotypes in malignant samples was 2/50 (4%) (Table 1): one in ductal carcinoma in situ (DCIS) and the other in invasive ductal carcinoma (IDC).

## 3. Discussion

A number of previous studies have clearly addressed the carcinogenic role of HPV as being a necessary cause to induce cancer in humans by transforming the virus-infected cells into a malignant phenotype [15,16]. Some studies have detected the presence of HPV in BC specimens and reported a wide range (4% to 86%) of viral prevalence [9,17]. The identified histological features of BC include the presence of epithelial hyperplasia, hypergranulosis, parakeratosis and koilocytes, which are consistent with HPV infections [13,18].

Investigating the mere presence of viral DNA in breast tissue is not enough to confirm a viral etiological role in BC development in humans. The HPV genome is now known to be integrated into the host genome [19,20]. HPV infection also causes direct effects through the expression of viral oncoproteins that influence carcinogenic mechanisms and its oncogenic role is well-known in cervical cancer [21]. It has recently been suggested that the HPV oncogenicity in breast tissue could be similar to cervical carcinogenesis, in which viral infection could be an early event, followed by cumulative changes over the chronic viral infection [22]. 

Several studies reporting the presence of HPV in breast cancer have linked the viral activity levels to the invasiveness of the disease. Consistent with these observations, our collaborator in the UK examined the expression of HPV oncoproteins in breast cancer tissue specimens, where the expression of HPV oncoprotein was observed in 19% of invasive breast cancer samples. These observations clearly give support to a potential role of HPV in breast cancer progression [14]. 

In the general population of Qatar, very limited data is available on HPV prevalence, incidence and genotype-specific dissemination. However, previously, HPV was identified in 18.4% of Arabic-origin women residing in Qatar with abnormal cervical cytology [23]. On the other hand, the presence of HPV was earlier documented to be as high as 64% in the same population [24]. It can be argued that cancer prevalence patterns are likely to have been altered along with lifestyle, cultural, environmental and social changes in Qatar. It is therefore reasonable that researchers should investigate additional factors that a general population living in rapidly changing societies like Qatar might face.

The primary aim of this study was to assess the prevalence of high-risk HPV infection in breast tissue in the prevailing population of Qatar. Thus, the presence of HR-HPV types was studied in 150 (50 malignant, 50 benign, 50 normal) fresh breast tissue samples. The presence of HR-HPV types that were detected in malignant and benign/normal samples were 10% and 8% respectively (Table 1).

The values identified in our study were lower than those described in a meta-analysis of 29 published studies, in which the prevalence rates of HPV infection were 23% in BC samples and 12.9% in normal cases. However, in contrast to the present study findings, studies conducted in France, Iran and Switzerland failed to detect the presence of HR-HPVs in benign breast tissue specimens of the related population [25,26,27].

It is interesting to observe that 80% (4/5) of malignant samples positive for HR-HPV were from invasive ductal carcinoma cases. This observation is in-line with previous studies which highlighted the highest HPV positivity rate in patients with ductal breast carcinoma as compared to other histological types. This may indicate two features of HPV infection. Firstly, the exposure of the mammary ducts to the external environment could possibly provide an entry point for HPV infection. Secondly, that HR-HPV infections may interact or act synergistically to initiate cancer development or increase the severity or risk of BC progression in the presence of other cofactors. 

Viral infection alone is not a sufficient cause for malignancy. Long-term virus persistence and the participation of other cofactors are needed to increase the risk of cancer progression [28,29]. In support of this statement, our data may suggest that the detection of HPV infection in normal breast specimens is expected to be an early event at the time of diagnosis, followed by progressive and cumulative changes over the years, similar to cervical carcinogenesis [22].

It is important to note that the prevalence of HPV-infection and HPV-type distribution in women within the general population varies significantly by country, regions within the same country, and population [30]. In addition, other risk factors such as genetic variation, sexual behavior, lifestyle and population immunity may influence the prevalence of HPV [31]. Consequently, part of this present investigation has been set to compare and evaluate the prevalence of 12 HPV types in breast cancer tissue in different populations, social behaviors and lifestyles in different geographical areas i.e., a cross-cultural comparison between the population of Qatar and the UK. With regards to HPV infection, we detected a great difference in prevalence amongst the two different populations in which the prevalence of HPV was 47% and 10% in UK and Qatar, respectively. Also, the most detected HPV types in breast cancer tissues were HPV 39, 18 and 45 in the UK population while the detected HPV types in the population of Qatar were HPV 16, 35 and 58 [14]. This explains the findings from the present study in a cohort of Qatari residents, where the researchers were able to detect viral DNA in five Filipino subjects (three malignant and two normal) in which HPV 35 and 16 were the most common types with a prevalence of 80% (4/5) and 40% (2/5) respectively (Table 2). Therefore, further study is needed to investigate potential factors that could explain this difference, prevalence and genotypes distribution of HPV within the investigated populations.

Additionally, we performed a retrospective investigation using medical records for Pap smear history to identify the presence of HPV types in the population of Qatar. We were able to retrieve the Pap smear results for 31 out of 150 subjects. No conclusions could be drawn from this finding since none of our HPV-positive subjects had been tested with a Pap smear apart from one subject with an unsatisfactory smear result. This highlights the necessity of implementing the national cervical screening program aiming to screen the entire eligible population in Qatar. 

There are some limitations to this study. The small sample size is probably the biggest limitation. This sample size is small for a meaningful comparative statistical analysis. Furthermore, the mere presence of viral DNA does not demonstrate an active infection. HPV protein expression is a known reliable marker to detect an active infection in breast tissue [14]. Therefore, further statistical analysis and studies with larger samples are required to determine if these HPVs were biologically active in these specimens to validate these findings. Finally, it would be of great interest to evaluate the correlation of HPV and its oncoproteins with clinicopathological parameters including hormone receptors, breast cancer stage and grade, in addition to patients’ age and sex.

## 4. Materials and Methods

### 4.1. Subjects, Enrolment and Breast Tissue Collection

The study was conducted in accordance with the Declaration of Helsinki and the protocol was formally approved by the ethics committee of the Institutional Review Board of Hamad Medical Corporation (HMC-IRB reference: 16101/16). All methods were carried out in accordance with the approved guidelines. Following the approval, all subjects gave their informed consent for inclusion before they participated in the study. A total of 150 subjects were enrolled and consented to obtain fresh breast tissue specimens (50 benign, 50 cancerous and 50 normal which served as control). The samples were aseptically collected by a dedicated surgical team at the Breast Surgery Department at Hamad General Hospital, Doha, Qatar. Normal and benign breast tissues were obtained from reduction mammoplasties, post-mastectomy contralateral balancing surgery and tissue removed from benign breast disease patients e.g., breast fibroadenomas. All cases were formally reported by the Pathology Department as part of the standard clinical diagnosis of the patients to be histopathologically classified into cancerous and non-cancerous cases (Table 1). 

This being a blind comparison study, histopathological information was masked from the researchers to prevent bias in research results. The collected tissue biopsies were coded and immediately preserved using Allprotect reagent from QIAGEN to stabilize the DNA, RNA and protein, with aseptic handling of tissue in the operating theatre environment to avoid sample contamination.

### 4.2. DNA extraction and Purification

An aseptic protocol was used during tissue handling to avoid cross-contamination between specimens. Consequently, separate disposable items, such as gloves, surgical blades and tubes were used for each sample. 

Genomic DNA was extracted from the breast tissue using a GenElute™ RNA/DNA/Protein Plus Purification Kit (Sigma) in accordance to the manufacturer’s protocol. In brief, residual Allprotect® tissue reagent was removed by dabbing and rolling the tissue sample over a sterile paper towel. Each sample was placed in a sterile Petri dish and carefully diced using a sharp sterile surgical blade under a clean sterile hood. The amount of tissue was accurately measured with a weight ranging between 20 and 30 mg and then homogenized in lysis buffer using gentle MACS™Octo Dissociator (MiltenyiBiotec, Bergisch Gladbach, Germany). The *h_tumor-01* predefined program was used to dissociate breast tissue at 37 °C. The lysate was loaded into a gDNA purification column to bind the DNA. The gDNA was washed and then eluted using an elution buffer. The quality and quantity of the extracted gDNA was assessed using a NanoDrop 2000c Spectrophotometer (Thermo Scientific, Waltham, Mass., USA). 

### 4.3. The Detection and Genotyping of HR-HPV

The identification of 12 HR-HPV types was performed by implementing the Polymerase Chain Reaction (PCR) technique using the HPV-HCR Genotype-Eph kit (AmpliSens, Bratislava, Slovak Republic). In order to avoid cross-contamination, DNA extraction and PCR technique were performed at two separate laboratories under a very strict aseptic protocol. Simultaneous amplification of four targeted regions of the HPV gene of four different HPV DNA types was run in one tube (multiplex-PCR). To amplify the 12 HR-HPV types simultaneously, three tubes were run, in which four different HR-HPV types were amplified in each tube as follows: HPV16/31/33/35, HPV18/39/45/59 and HPV52/56/58/66. This approach enabled the detection of infections and co-infections of 12 HR-HPV types in each sample. 

The amplified PCR products were electrophoresed along with type-specific positive controls and negative control on a 3% (w/v) agarose gel stained using SYBR Safe (Invitrogen, Carlsbad, CA, USA). The quality of the DNA obtained from each sample was controlled by amplification with primers to detect the β-globin gene. The gel was then visualized under ultraviolet (UV) transilluminator using a Gel Doc XR+ System (Bio-Rad, Hercules, CA, USA). Since PCR products differ in length, the genotype of HR-HPV was identified accordingly.

## 5. Conclusions

This study investigated the presence of 12 HR-HPV DNA in fresh breast tissue samples in the general population of Qatar through prospectively collected data. The viral DNA was detected in 10% of breast cancer samples. However, the virus was also detected in 8% of benign/normal breast samples. Though the viral detection is not significant enough to validate HPV as a risk factor for breast cancer in the population of Qatar, our data is in agreement with previous studies which detected a low prevalence of HPV DNA in breast cancer.

## Figures and Tables

**Figure 1 cancers-12-01528-f001:**
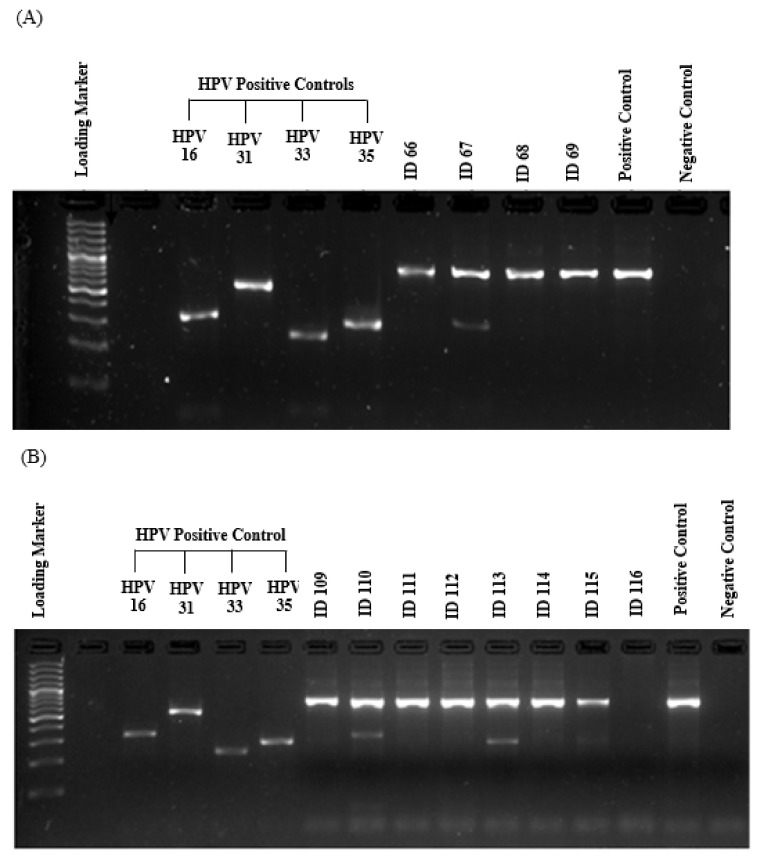
Gel electrophoresis pattern of HR-HPV types (16/31/33 and 35). (**A**) and (**B**) are representative gel electrophoresis patterns of the analysis for HR-HPV types of breast tissue. Samples: (**A**) loading marker = 100 bp plus DNA ladder (100 bp–3000 bp), HPV positive controls = HPV16 (325 bp), HPV31 (520 bp), HPV33 (227 bp) and HPV35 (280 bp) respectively; ID66, 68 and 69 = HPV negative clinical sample, ID67 = HPV positive clinical sample, positive control (C+) = internal control, human DNA (β-globin 723 bp), negative control (C-); (**B**) loading marker = 100 bp plus DNA ladder (100 bp–3000 bp), HPV positive controls = HPV16 (325 bp), HPV31 (520 bp), HPV33 (227 bp) and HPV35 (280 bp) respectively; ID109, 111, 112, 114, and 115 = HPV negative clinical sample, ID110, 113 = HPV positive clinical sample, ID116 = absent internal control (β-globin 723 bp) indicating inadequate quality of sample; positive control (C+) = internal control, human DNA (β-globin 723 bp), negative control (C-).

**Figure 2 cancers-12-01528-f002:**
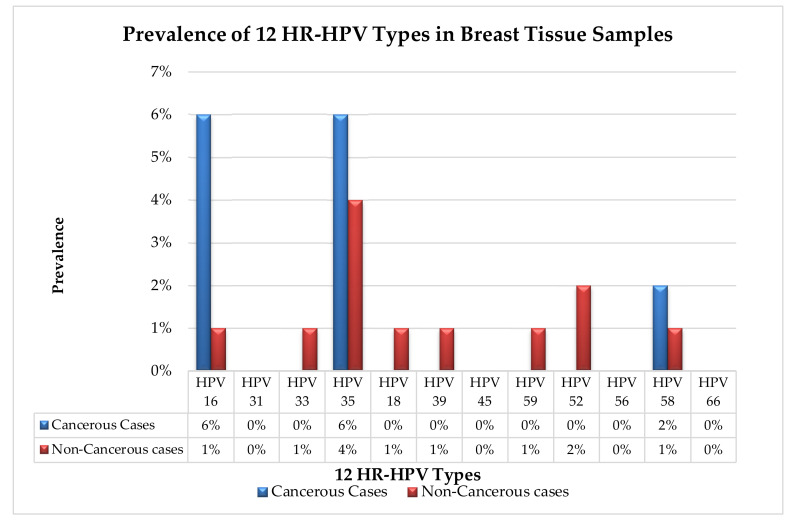
Prevalence of 12 HR-HPV types in breast tissue samples. The graph illustrates the prevalence of each of the 12 HR-HPV types in cancerous and non-cancerous breast tissue. The most prevalent HPV types in the cancerous cases were HPV 16 and 35 with a prevalence of 6% followed by HPV 58 with a prevalence of 2%. The most prevalent HPV type in non-cancerous cases was HPV 35 with a prevalence of 4% followed by HPV 52 with a prevalence of 2% and HPV 16, 33, 18, 39, 59 and 58 with a prevalence of 1%.

**Table 1 cancers-12-01528-t001:** Identification and frequency of high-risk human papillomavirus (HR-HPV) infection in cancerous, benign and normal breast tissue specimens.

	Total Samples *N* (%)	HPV + ve *N* (%)		
Total Number of Samples *N*	150 (100)	13/150 (8.7)		
Male Female	10 140			
Age (Year) (15–84)				
*<50*	116/150 (77)			
*>50*	34/150 (23)			
Pathological Status			Single HPV Infection *N* (%)	HPV Co-Infection *N* (%)
Cancerous Cases	50/50 (100)	5/50 (10)	3/50 (6)	2/50 (4)
In Situ Cases				
Ductal Carcinoma in Situ (DCIS)	15/50 (30)	1/15 (6.6)	-	1/2 (50)
Lobular Carcinoma in Situ (LCIS)	0/50 (0)	-	-	-
Invasive Cases				
Invasive Ductal Carcinoma (IDC)	33/50 (66)	4/33 (12.12)	3/3 (100)	1/2 (50)
Invasive Lobular Carcinoma (ILC)	1/50 (2)	-	-	-
Invasive and In Situ Cases				
Invasive and In Situ Ductal Carcinoma	0/50 (0)	-	-	-
Invasive and In Situ Lobular Carcinoma	1/50 (2)	-	-	-
Non- Cancerous Cases	100/100 (100)	8/100 (8)	6/100 (6)	2/100 (2)
Benign Fibroadenoma	23/100 (23)	3/23 (13)	3/6 (50)	-
Benign Phyllodes Tumor	4/100 (4)	1/4 (25)	1/6 (16.7)	-
Benign Breast Tissue	13/100 (13)	-	-	-
Gynecomastia	5/100 (5)	-	-	-
Papillomatosis	5/100 (5)	-	-	-
Normal	50/100 (50)	4/50 (8)	2/6 (33.3)	2/2 (100)

- negative.

**Table 2 cancers-12-01528-t002:** HPV Frequency in Positive Samples. DCIS: ductal carcinoma in situ; IDC: invasive ductal carcinoma.

ID	Pap Smear Status	HPV Status	Breast Histopathology Status	Nationality
18	0	HPV 58	Benign/Fibroadenoma	Indian
19	0	HPV 16 and 58	Malignant/IDC	British
63	0	HPV 35	Malignant/IDC	Afghani
65	0	HPV 59	Benign/fibrous histiocytoma	Filipino
67	0	HPV 35	Benign/phyllodes tumor	Filipino
104	0	HPV 35	Malignant/IDC	Syrian
107	0	HPV 16 and 35	Malignant/DCIS	Filipino
110	0	HPV 16	Malignant/IDC	Yemeni
113	0	HPV 35	Benign/ Fibroadenoma	Ethiopian
129	0	HPV 33 and 35	Normal	Filipino
130	0	HPV 39	Normal	Qatari
139	*	HPV 52	Normal	Lebanese
145	0	HPV 16, 35, 18, 52	Normal	Filipino

* unsatisfactory.

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
