# Peer review of "Prevalence and Type Distribution of High-Risk Human Papillomavirus (HPV) in Breast Cancer: A Qatar Based Study"

_cancers, 2020, doi:10.3390/cancers12061528_

Round 1

Reviewer 1 Report

I have completed my revision of the manuscript; it is well designed, the relevance of the topic is fair and results confirm the data already in literature. the authors state that the hypotesis that HPV could cause Breast Cancer is not corroborated by the data showed in Results. The little percentage of viral dna in breast cancer is the same we have observed in previous trials. Materials and Methods Section is located after Discussion it's not a real problem but unusual. The manuscript can be published after minor revision without any priority.

Author Response

Please see the attached file for reviewer 1 response

Reviewer 2 Report

Dear Editor

I have read the manuscript entitled: " Prevalence and Type Distribution of High Risk Human Papillomavirus (HPV) in Breast Cancer: A Qatar based study"

This is a well-designed study. The PCR tests to detect the different HPVs are correct. But no statistical test has been done. I believe the authors are able to calculate an Odds Ratio and its confidence intervals, namely OR 1.2778 95% CI 0.3954-4.1289 significance level 0.6821.

On the other hand, I have a doubt: how could they have obtained normal tissue without violating the Declaration of Helsinki? Healthy tissue can not be removed from a breast if it does not provide a possible benefit to the patient. I can only think of a breast reduction. I think this point should be clarified by the authors. This point is the clue of my decision, major revision.

Another aspect to point out is the lack of a paragraph on limitations. for example, there is no further analysis of whether the two samples are comparable by age or sex (since the study involves 10 men).

And finally, they have not cited one of the best baseline studies linking breast cancer and HPV, with very different results: Delgado-García S, Martínez-Escoriza J-C, Alba A, Martín-Bayón T-A, Ballester-Galiana H, Peiró G, et al. Presence of human papillomavirus DNA in breast cancer: a Spanish case-control study. BMC Cancer [Internet]. 2017;17(1):320, available from http://bmccancer.biomedcentral.com/articles/10.1186/s12885-017-3308-3

Kind regards

Author Response

Please see the attached file for reviewer 2 reponse

Round 2

Reviewer 2 Report

No Comments

Author Response

We appreciated reviewer for no further comments